

# How does physical activity improve adolescent resilience? Serial indirect effects *via* self-efficacy and basic psychological needs

Xuening Li[1,*], Jing Wang[2,*], Huasen Yu[3], Yang Liu[4], Xiaoling Xu[5], Jiabin Lin[6] and Ning Yang[7]

[1] Brain and Cognition Research Center (CerCo), Université Paul Sabatier (Toulouse III), Toulouse, CHU Purpan, France
[2] School of Physical Education, Northeast Normal University, Changchun, China
[3] College of Physical Education and Health, East China Normal University, Shanghai, China
[4] School of Physical Education, Shaanxi Normal University, Xi'an, China
[5] Fujian Province Nanping No. 1 High School, Nanping, China
[6] School of Physical Education, Changchun Normal University, Changchun, China
[7] Institute of Physical Education, Shandong Youth University of Political Science, Ji'nan, China
[*] These authors contributed equally to this work.

Corresponding authors
Xuening Li, lixn563@nenu.edu.cn
Ning Yang, 122453656@qq.com

## ABSTRACT

**Background**. Resilience is vital for improving mental health and well-being during adolescence, which is an important yet vulnerable period. Previous research has indicated that physical activity enhances individual resilience. However, limited studies have examined underlying psychological mechanisms between them. The current study aimed to investigate the effect of physical activity on adolescent resilience *via* self-efficacy and basic psychological needs.

**Methods**. A cross-sectional survey was conducted with 1,732 high school students aged 16 to 20 years old (mean age: $16.51 \pm 0.77$ years), with nearly equal number of boys (47.63%) and girls (52.37%). They each completed the Physical Exercise Questionnaire, Basic Psychological Needs in Exercise Scale, General Self-Efficacy Scale, and Resilience Scale, respectively. A serial indirect model was constructed to examine how physical activity influences resilience.

**Results**. Structural equation model analysis revealed that physical activity significantly and directly predicted resilience. When self-efficacy and basic psychological needs were included in the model, both direct and indirect effects were observed. Specifically, the positive relationship between physical activity and resilience was partially mediated by self-efficacy and basic psychological needs. In addition, basic psychological needs and self-efficacy were found to serially mediate the direct relathonship between physical activity and resilience.

**Conclusions**. The present study provides novel theoretical insights into sports psychology by establishing a link between basic psychological needs and self-efficacy. The findings have implications for school administrators and physical education instructors in designing targeted interventions to promote adolescent resilience. These interventions may involve creating supportive environment conductive to fulfilling students' basic psychological needs, implementing strategies to enhance self-efficacy

beliefs, and providing opportunities for skill development and mastery experiences in sports and physical activities.

## INTRODUCTION

Under challenging conditions such as interpersonal tension and intense academic pressure, adolescents are vulnerable to the harmful effects of negative events (*Yang et al., 2022*), which leads to a notable increase in anxiety, depression, stress levels, and feelings of loneliness (*Lee, 2020*; *Nkire et al., 2021*). However, as *Li et al. (2021)* noted, despite many individuals facing the same ongoing challenges and crises, there are still variations among individual responses. This raises the question of why some adolescents are capable of recovering from and adapting to these challenges, while others are not. We suggest that certain students possess greater personal resilience, enabling them to more adeptly manage adverse events (*Rayburn, Anderson & Sierra, 2021*). In other words, resilience plays an important role in explaining such variance. It is widely recognized that personal resilience can be developed, reinforced, and sustained through physical activity (*Ho et al., 2015*). Despite the abundance of research suggesting a positive relationship between physical activity and adolescent resilience, our understanding of the underlying psychological mechanisms remains limited. Thus, examining how physical activity promotes teen resilience is vital. Recent studies have shown that the relationship between physical activity and resilience can be mediated by basic psychological needs and self-efficacy (*Neumann et al., 2022*; *Xu et al., 2021*). We aim to address this issue *via* a cross-sectional survey of high students investigating whether physical activity improves resilience by enhancing basic psychological needs and self-efficacy.

Resilience is a trait that enables an individual to adjust and adapt to life's adversities or risks (*Van der Merwe, Botha & Joubert, 2020*). It also reflects the ability of an individual to adapt, recover, and maintain a positive outlook in the face of adversity, stress, and challenges (*Hughes et al., 2021*; *Jackson, Firtko & Edenborough, 2007*). It is not a fixed trait but rather a dynamic process that can be developed and strengthened over time through various coping strategies and protective factors (*Kalisch, Müller & Tüscher, 2005*). Physical activity plays an important role in building resilience (*Carriedo et al., 2020*; *Ozkara et al., 2016*). Individuals who engage in high levels of physical activity are more likely to develop high levels of resilience (*Ho et al., 2015*). Behavioral evidence supports the notion that physical activity has a beneficial influence on resilience by enhancing positive psychological states, optimizing stress reactivity, and protecting against the effects of stressful events (*Arida & Teixeira-Machado, 2021*; *Li et al., 2021*). Neuroimaging studies have also provided important evidence about the tight link between physical activity and resilience. Research has suggested that physical activity can facilitate resilience by promoting brain structure and function in neural circuits involved in self-regulation (for a review, see *Belcher et al.,*

*2021*). For example, *Dahl et al. (2017)* reported that regular physical activity may positively affect prefrontal cortex brain structure and function by regulating bottom-up processing, thus strengthening resilience. In particular, aerobic exercise has the potential to influence the intrinsic functional organization of large-scale networks, such as cognitive control and motor networks, improving emotional and behavioral regulation (*Krafft et al., 2014*), thereby improving resilience. In summary, evidence indicates that physical activity is a key strategy for building resilience in adolescents.

It is of utmost importance to have a clear understanding of the mechanisms that underlie the positive impact of physical activity on resilience. The existing literature suggests that self-efficacy is not only a determinate but also an outcome of physical activity (*Martin et al., 2015*). Self-efficacy, as defined by *Bandura (1977)*, refers to an individual's belief in his or her ability to successfully perform a particular task. This finding indicates that individuals with high self-efficacy are more likely to have greater confidence in completing difficult tasks, are more willing to exert effort, and exhibit greater persistence in the face of adversity. Many studies have demonstrated that adolescents who regularly participate in physical activity exhibit a higher level of self-efficacy (*Dishman et al., 2006*; *Wu et al., 2022*). In particular, a study by *Netz et al. (2005)* found that moderate-intensity aerobic training had the greatest impact on improving self-efficacy levels in healthy adults. *Zhao & Ma*'s (*2019*) cross-sectional study revealed a significant positive correlation between physical activity and self-efficacy for high school students. Within the framework of the Positive Appraisal Style Theory of Resilience by *Kalisch, Müller & Tüscher (2005)*, self-efficacy expectations have been found to have a beneficial association with stress resilience. A study performed by *Sánchez-Teruel & Robles-Bello (2020)* among young men in Morocco revealed that self-efficacy is one of the most predictable and positive factors affecting resilience. Thus, resilience and self-efficacy are closely intertwined. Further, *McAuley et al. (2011)* conducted a review and provided strong evidence that self-efficacy could mediate the association between physical activity and psychological outcomes. Taken together, these studies strongly support that self-efficacy plays a role in the buffering effect of physical activity and in how physical activity contributes to resilience.

Basic psychological needs are a core concept in self-determination theory (SDT). According to the SDT framework, basic psychological needs include autonomy, competence, and relatedness. These needs are essential for human thriving and wellbeing, as they serve as universal sources of nourishment for individuals (*Ryan & Deci, 2000*; *Ryan & Deci, 2017*). *Wilson et al. (2008)* suggested that the theory of basic psychological needs provides a promising framework for studying students' psychological health and behavior and is important for promoting positive and healthy development in physical activity contexts. When students who actively participate in sports experience higher levels of relatedness need satisfaction than do non-sports active students during a physical education course (*Burchard Erdvik et al., 2019*). Physical activity is positively and significantly associated with basic psychological needs among adolescents (*Xu et al., 2021*). The greater their competence is, the greater their satisfaction with exercise (*Wilson et al., 2006*). A longitudinal study of Iranian students revealed that a 10-day physical activity intervention significantly increased the experience of basic psychological needs and

decreased stress (*Behzadnia & FatahModares, 2020*). Furthermore, previous works have shown that greater resilience is correlated with better basic psychological needs (*e.g.*, *Lera & Abualkibash, 2022*). Relatedness need may have the most significant effect (*Naemi, 2018*). Another study revealed that satisfaction of competence, but not autonomy or relatedness, was positively related to increased resilience (*Neufeld & Malin, 2019*). Overall, these streams of research provide a better understanding and valuable insights into aspects of basic psychological needs as potential mediators of in the relationship between physical activity and an individual's resilience.

Interestingly, the extant literature highlights the association between basic psychological needs and self-efficacy among adolescents (*Vayskarami et al., 2019*; *Zhen et al., 2017*). Conceptually, competence is a core component of basic psychological needs and self-efficacy, as it refers to an individual's self-perceptions or self-judgment of their skills and capabilities. When an individual's competence needs are satisfied, they are more likely to experience a stronger sense of self-efficacy. Specifically, the satisfaction of competence needs in an exercise process may lead individuals to feel that they have the ability to participate in and complete various exercise activities, resulting in a higher perception of physical self-efficacy (*Zhang & Xi, 2010*). *Macakova & Wood (2022)* found that both competence and relatedness needs were positively associated with self-efficacy. This finding suggests that individuals who have their competence and relatedness needs met are more likely to have a stronger sense of self-efficacy. Moreover, when an individual's relatedness needs are fulfilled, it can facilitate the development of a robust and optimal social network, which can better allow them to cope with adversities through effective support mechanisms (*Nawaz & Gilani, 2011*) and will contribute to stronger self-efficacy. The fulfillment of autonomy can signify not only an individual's perception of their own competency but also their ability to regulate their behavior based on their own will, which can enhance their self-efficacy (*Ryan & Deci, 2000*). Specifically, autonomy, competence, and relatedness needs have been found to be significant predictors of self-efficacy. As previously mentioned, basic psychological needs and self-efficacy may serve as factors contributing to the indirect effects of physical activity on resilience.

As mentioned above, despite previous studies highlighting positive relationships among physical exercise, basic psychological needs, self-efficacy, and resilience, there has been a lack of examination of these variables as interactive systems. Furthermore, existing research on the relationship between physical activity and resilience has focused primarily on college students (*Xu et al., 2021*; *Zhang et al., 2022*). Thus, we aimed to investigate the psychological mechanism of the effect of physical activity on resilience in high school students. Our research provides theoretical evidence for understanding the relationship among high school students and developing targeted interventions or programs designed to enhance adolescent resilience. To this end, we propose the following hypotheses: (H1) there is a direct positive association between physical activity and resilience; (H2) physical activity has an indirect effect on resilience through satisfying basic psychological needs; (H3) physical activity has an indirect effect on resilience through improving the levels of self-efficacy; and (H4) physical activity improves resilience through serial indirect effects of basic psychological needs and self-efficacy.

## METHODS

### Study design

The current investigation employed a quantitative cross-sectional design implemented on the 'Wenjuanxing' platform to explore the direct and indirect associations between physical activity (as the independent variable) and resilience (as the dependent variable). Additionally, this study examined whether the association could be influenced by self-efficacy and basic psychological needs using a serial indirect effect model.

### Participants

The participants completed an online questionnaire at their respective school. A total of 1,732 students were included in the final sample after excluding 28 due to regular answering patterns or faulty data, resulting in a valid response rate of 98.35%. The participants were in their second year of high school and ranged in age from 16 to 20 years old ($M = 16.51$; $SD = 0.77$), with nearly equal number of boys (47.63%) and girls (52.37%). All six included schools approached agreed to participate, and the study was approved by the Academic Committee of the School of Shandong Youth University of Political Science and is also in accordance with the latest revised ethical guidelines of the Declaration of Helsinki.

### Procedures

We utilized a simplified cluster sampling method to randomly select participants for our study. This method is a two-step process involving dividing the entire population into clusters or groups, such as geographic areas or districts (*e.g.*, villages, schools, wards, blocks) (*Acharya et al., 2013*). In our study, we first used a two-cluster design, consisting of southern and northern regions of China. We then randomly selected three high schools from each cluster, resulting in a total of six selected schools. From each of these schools, a certain number of students were then randomly selected from each school. In addition, we selected a wide range of centers to recruit participants, including both public and private. A total of 1,760 students were selected for our study over a two-week period in Sept. 2022. Written informed consent was obtained from all participants, who were provided a detailed introduction to the study and its purpose, as well as declarations of anonymity and confidentiality before their participation.

### Measure

Physical activity was assessed using the Physical Activity Questionnaire (*Wu, 2016*). The author of the questionnaire has made it publicly available and free for use. The Physical Activity Questionnaire has been widely used to assess physical activity in Chinese adolescents (*Li, Yu & Yang, 2021*), and it consists of eight items for assessing exercise adherence (*i.e.,* 4-itemsp; *e.g.*, "It is difficult for me to quit physical activity") and exercise commitment (*i.e.,* 4-itemsp; *e.g.*, "I have the habit of exercising"). Each item is scored on a 5-point Likert scale ranging from 1 (totally disagree) to 5 (totally agree). The total scores range from 0 to 40, with higher scores indicating a greater amount of exercise. The Cronbach's alpha of this measure in the present research was 0.85, ranging from 0.74 to 0.82 across the two subscales.

The Basic Psychological Needs in Exercise Scale (BPNES), developed by *Vlachopoulos & Michailidou (2006)*, was used to assess basic psychological needs. The Chinese version, which was previously translated by *Liu, Chung & Duan (2013)*, was used in this study. Additionally, *Liu, Chung & Duan (2013)* demonstrated that the Chinese version had good validity. This scale contains three subscales, competence, relatedness, and autonomy, and consists of 12 items. Participants responded to each item on a 7-point Likert scale ranging from 1 (strongly disagree) to 7 (strongly agree). Higher scores indicate higher levels of need satisfaction. The Cronbach's alpha for the overall BPNES was 0.93 in this study, and for each of the three subscales, it ranged from 0.79 to 0.92.

The General Self-Efficacy Scale (GSES) was adopted to measure individual self-efficacy. Specifically, we used a Chinese version of the GSES, which has been validated for use with Chinese adolescents (*Zhang & Schwarzer, 1995*). The Chinese version was authorized and approved by the original author (*Schwarzer, 1993*) and further demonstrated to possess greater validity (*Wang, Hu & Liu, 2001*). The scale consists of 10 items rated on a 4-point Likert scale, ranging from 1 (not at all sure) to 4 (completely true). The total scores range from 10 to 40 points, with higher scores representing greater self-efficacy. The Cronbach's alpha for the present study was 0.93.

A modified Chinese version of the Connor-Davidson Resilience Scale was used in Chinese adults to assess resilience levels (*Yu & Zhang, 2007*; *Li et al., 2021*). All the translations and adaptations of the Chinese version were authorized and approved by the original author (*Connor & Davidson, 2003*). The scale contains 25 items that evaluate tenacity (13 items), strength (eight items), and optimism (four items). Each item is scored on a 5-point Likert scale from 0 (never) to 4 (always), and the total scores ranged from 0 to 100 points, with higher scores indicating greater resilience. However, one item ("sometimes fate and God can help") on the optimistic scale had a factor loading of less than 0.40 (*Ford, MacCallum & Tait, 1986*) and was removed from the analysis. The Cronbach's alpha for overall resilience was 0.95, and for each of the three subscales, it ranged from 0.66 to 0.92, indicating high internal consistency.

## Data analysis

The full path model, which included physical activity level, resilience, basic psychological needs, and self-efficacy, was used for the power analysis. The complete model included four degrees of freedom and needed a sample size of 1,194 to identify a close fit with an RMSEA value of 0.05 and 80% statistical power (*Kim, 2005*). A total of 1,732 subjects were included in this study, providing sufficient statistical power.

The results of the present study were analyzed using a two-step process. First, all the preliminary data analyses were performed using SPSS 21.0. This included calculating descriptive statistics such as the means and standard deviations for the social demographic and the variables to summarize the basic features of the data. Reliability analysis was subsequently performed to explore the internal consistency of the measurement scales. Pearson correlation analysis was then used to measure bivariate relationships among physical exercise, resilience, self-efficacy, and basic psychological needs. Finally, a hypothesized serial multiple mediator model was constructed using Amos 24.0 to investigate

**Table 1 Descriptive statistics and Pearson correlation coefficient among research variables ($N = 1,732$).**

| Variables | 1 | 2 | 3 | 4 | 5 | Mean | SD |
|---|---|---|---|---|---|---|---|
| 1.Sex | 1 | – | – | – | – | – | – |
| 2.Physical activity | −014 | 1 | – | – | – | 24.04 | 5.18 |
| 3.Self-efficacy | −016[**] | 0.40[**] | 1 | – | – | 25.89 | 6.00 |
| 4.Basic phycological needs | −011[**] | 0.34[**] | 0.56[**] | 1 | – | 62.84 | 12.77 |
| 5.Resilience | −012[**] | 0.50[**] | 0.70[**] | 0.65[**] | 1 | 62.37 | 16.46 |

Notes.
[**]$p < .01$.

the direct and indirect effects of physical exercise on resilience. Goodness-of-fit indices were used to assess the fit of the serial multiple mediator model. Specifically, the chi-square ($\chi^2$) test directly evaluates how well the proposed model fits the data (*Bollen, 1989*). A $\chi^2$ result of $p < 0.05$ is significant and indicates that the proposed model does not fit the data. Conversely, a nonsignificant $\chi^2$ result ($p > 0.05$) indicates that the proposed model fits the data well (*Barbeau et al., 2019*; *Kim & Faith, 2020*; *Shah, 2012*). The root mean square error of approximation (RMSEA) was also used to evaluate whether the proposed model supported factor structure (*Fung et al., 2020*; *Lin et al., 2021*; *Schweizer, 2010*). A value less than 0.05 indicates a close model fit, while values between 0.05 and 0.08 indicate an acceptable model fit (*Barbeau et al., 2019*; *Browne & Cudeck, 1989*). The comparative fit index (CFI), normal fit index (NFI), goodness-of-fit index (GFI), and adjusted goodness-of-fit index (AGFI) were also used as important measurement indices; for each, a value higher than 0.90 was regarded as an acceptable model fit (*Bentler, 1990*). Overall, the fit of the proposed model was determined using the chi-square RMSEA, CFI, NFI, GFI, and AGFI. Additionally, a bootstrapping analysis with 5,000 random resamples was carried out to test the significance of the mediating effects and estimate 95% bias-corrected bootstrap confidence intervals. The level of statistical significance for all indicators was set at $p < 0.05$.

## RESULTS

### Descriptive statistics and preliminary analyses

It is important to address the potential issue of common method variance, which arises from using self-reported data from the same source (*Podsakoff et al., 2003*). We employed Harman's single factor test utilizing nonrotated factor resolution (*Podsakoff et al., 2003*). The first factor alone explained only 37.29% of the variance in the data. Hence, we can confidently state that common method variance does not pose a substantial threat in this study.

Table 1 presents the mean, standard deviation, and Pearson and point-biserial correlations of the variables. With the exception of certain correlations related to sex, all of the main variables in the study were correlated, ranging from −0.34 to 0.70. Overall, the results suggest that physical activity, self-efficacy, and basic psychological needs have a considerable influence on resilience.

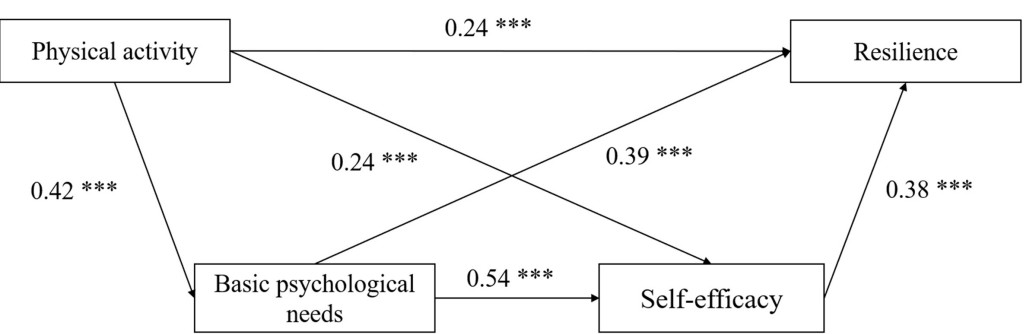

**Figure 1 A serial indirect effect model for physical activity, basic psychological needs, self-efficacy, and resilience.**

## Mediation analysis

A structural equation model was established using Amos software to explore the serial mediating effect of physical exercise on adolescents' resilience between basic psychological needs and self-efficacy. The model's fitting indices were extremely good, with $\chi^2/df = 6.85$, RMSEA = 0.06, CFI = 0.98, NFI = 0.98, GFI = 0.98, and AGFI = 0.96. In the serial mediation analysis, all the direct effects estimated from path analysis are shown in Fig. 1. According to this model, and independent of sex, the total effect of physical activity on resilience was found to be significant ($\beta = 0.59$, SE = 0.03, $p < 0.01$, 95% CI [0.54–0.64]). There was a statistically significant direct association between physical activity and resilience ($\beta = 0.24$, SE = 0.03, $p < 0.01$, 95% CI [0.19–0.30]), which supports Hypothesis 1. Similarly, physical activity had a strong positive association with adolescents' basic psychological needs ($\beta = 0.42$, SE = 0.03, $p < 0.01$, 95% CI [0.37–0.48]) and self-efficacy ($\beta = 0.24$, SE = 0.03, $p < 0.01$, 95% CI [0.18–0.30]). There were also significant direct paths from basic psychological needs to resilience ($\beta = 0.39$, SE = 0.03, $p < 0.01$, 95% CI [0.32–0.45]) and from self-efficacy to resilience ($\beta = 0.38$, SE = 0.03, $p < 0.1$, 95% CI [0.32–0.45]). In a similar vein, basic psychological needs had a significant positive association with self-efficacy ($\beta = 0.54$, SE = 0.03, $p < 0.01$, 95% CI [0.48–0.59]).

Next, the significant indirect effects of all paths of physical activity on resilience are presented in Table 2. First, higher levels of physical activity were associated with higher levels of basic psychological need satisfaction and, in turn, greater resilience ($\beta = 0.16$, SE = 0.02, $p < 0.001$, 95% CI [0.13–0.20]); thus, Hypothesis 2 was supported. Second, a noteworthy indirect pathway was observed, indicating that physical activity exerted a significant influence on resilience solely through self-efficacy ($\beta = 0.09$, SE = 0.02, $p < 0.01$, 95% CI [0.07–0.13]) and supporting Hypothesis 3. Finally, there was a significant indirect pathway for physical activity through basic psychological needs and self-efficacy in a serial fashion ($\beta = 0.09$, SE = 0.02, $p < 0.01$, 95% CI [0.07–0.13]); great levels of physical activity were linked to the fulfillment of basic psychological needs and higher levels of self-efficacy and, in turn, ultimately contributing to the bolstering of resilience. Hence, Hypothesis 4 was also supported.

**Table 2  Toal, direct and indirect effects between physical activity and resilience ($N = 1,732$).**

| Relationship | Effect | Boot SE | 95% CI | |
|---|---|---|---|---|
| | | | Low | High |
| Total effects | | | | |
| Physical exercise → Resilience | 0.59 | 0.03 | 0.54 | 0.64 |
| Direct effect | | | | |
| Physical exercise → Resilience | 0.24 | 0.03 | 0.19 | 0.30 |
| Indirect effects | | | | |
| Physical exercise → Basic psychological needs → Resilience | 0.16 | 0.02 | 0.13 | 0.20 |
| Physical exercise → Self-efficacy → Resilience | 0.09 | 0.02 | 0.07 | 0.13 |
| Physical exercise → Basic psychological needs → Self-efficacy → Resilience | 0.09 | 0.01 | 0.07 | 0.11 |

**Notes.**
SE, standard error; CI, confidence interval.

# DISCUSSION

This investigation aimed to explore the direct and indirect effects of physical exercise on resilience. Our results suggest that the effect of physical activity is associated with resilience through the following mechanisms: (i) indirectly by satisfying of basic psychological needs; (ii) indirectly *via* higher levels of self-efficacy; and (iii) indirectly *via* the fulfillment of basic psychological needs and, sequentially, increased self-efficacy. As expected, physical activity was related to resilience but also indirectly related to basic psychological needs, and indirectly related to self-efficacy (*Neumann et al., 2022*; *Xu et al., 2021*). Additionally, engaging in physical activity was shown to contribute to increased levels of basic psychological needs, subsequently leading to greater self-efficacy and ultimately higher levels of resilience.

Our results are consistent with earlier findings suggesting that physical activity serves as a resilience factor among adolescents (*Dunston et al., 2022*; *San Román-Mata et al., 2020*). On the one hand, aerobic exercise is a potential coping resource for boosting adolescents' resilience by enhancing cognitive function and mental health (*Chen et al., 2017*) and positively influencing mood states such as depression and anxiety (*Eöry et al., 2021*). Thus, adolescents can obtain a sense of psychological and physical pleasure and satisfaction through physical activity, leading to positive changes in psychological functions (*Li et al., 2021*). On the other hand, the neuroplastic effects induced by physical activity help develop adolescents' resilience by strengthening individual brain regions and large-scale neural circuits (*Belcher et al., 2021*). These changes contribute to emotional self-regulation and may help mitigate the risk of emotional heathy problems during this critical yet vulnerable period. In summary, while it is clear that physical activity has direct and beneficial effects on resilience, the psychological and neural mechanisms underlying this relationship are complex and diverse.

As expected, across bivariate and multivariate analyses and across indirect models, basic psychological needs were a significant mediator linking physical activity and resilience. Our findings align with those of previous studies demonstrating that satisfying basic psychological needs in the context of physical activity can significantly predict resilience in

adolescents (*Franco & Coterón, 2017*; *Leversen et al., 2012*). On the one hand, physical activity correlated positively with basic psychological needs, as reported in *Franco & Coterón*'s study (*2017*). One possible explanation is that physical activity provides individuals with a sense of autonomy or control over their bodies and their lives (*Teixeira et al., 2012*). Engaging in physical activity allows individuals to choose when, where and how they exercise, which can increase feelings of self-determination and competence. When individuals improve their exercise levels, learn sports skills, or reach exercise goals, it can provide a sense of accomplishment and progress, thereby increasing their feelings of self-determination and competence. On the other hand, basic psychological needs may promote well-being and strengthen inner resources related to resilience; this positive influence helps individuals develop the resilience they need. Specifically, the satisfaction of relatedness needs promotes an individual's perception of self-value and importance (*Xu et al., 2021*). When the autonomy needs are met, individuals can perceive an enhanced sense of control over their environment (*Ntoumanis, Edmunds & Duda, 2009*). Individuals whose competence needs are satisfied are more inclined to experience intrinsic motivation and perseverance, ultimately enabling them to sustain a high level of effort even in the face of adversity (*Grolnick, Deci & Ryan, 1997*).

In a similar vein, our data revealed that engaging in physical activity significantly predicts self-efficacy, which in turn predicts adolescent resilience. These findings are consistent with our predictions and prior research that physical activity can serve as an efficacy experience in the physical domain, thereby promoting psychological wellbeing (*McAuley et al., 2005*). Engaging in physical activity provides individuals with opportunities to acquire skills, overcome challenges, and accomplish desired tasks. These mastery experiences contribute to an increase in self-efficacy (*Bandura, 2000*; *Bautista, 2011*). Regular physical activity is associated with various physiological and psychological benefits, such as increasing self-esteem, enhancing mood, reducing stress, and improving cognitive function (*Mahindru, Patil & Agrawal, 2023*). Experiencing these positive changes reinforces individuals' belief in their capabilities and contributes to a sense of self-efficacy. On the other hand, self-efficacy beliefs are instrumental in promoting resilience in the face of adversity by activating affective, motivational, and behavioral mechanisms. Consequently, self-efficacy is often conceptualized as an essential component of resilience. The current study further strengthens the evidence that self-efficacy serves as a crucial psychological mechanism underlying the role of physical activity in enhancing resilience.

Previous research has shown that fulfilling basic psychological needs is a significant predictive factor of self-efficacy (*e.g.*, *Cho et al., 2015*). Our results align with previous research revealing significant associations between basic psychological needs and self-efficacy. Therefore, the positive association between physical activity and resilience was also explained through the fulfillment of basic psychological needs in conjunction with the increase in self-efficacy. SDT posits that satisfying basic psychological needs for autonomy, competence, and relatedness is an innate drive and essential for ongoing psychological growth, internalization, and well-being (*Van den Broeck et al., 2016*), which, in turn, predicts more situation-specific self-efficacy (*Diseth, Danielsen & Samdal, 2012*). Moreover, the empirical research conducted by *Sari et al. (2011)* revealed that the

satisfaction of basic psychological needs significantly contributed to the perception of self-efficacy among students. To summarize, this paper provides an initial confirmation and expansion of evidence supporting the concept of serial mediation between physical activity and students' resilience.

These findings provide strong support for the positive correlation between physical activity and resilience. Consequently, educators can design interventions promoting physical activity programs tailored to individuals' preferences and psychological needs. By cultivating an environment that encourages autonomy, relatedness, and competence, individuals are more likely to willingly engaged in physical activities, thereby enhancing their resilience. Additionally, therapists and counselors can incorporate physical activity into therapeutic interventions, recognizing its potential to not only improve physical health but also bolster emotional resilience. By integrating physical activities that align with individuals' interests and psychological needs, therapy sessions can become more comprehensive and effective, promoting both physical and mental well-being.

Like other studies, this research has several limitations that should be acknowledged. First, although significant and positive relationships between various variables were observed, the cross-sectional design precluded establishing the exact causal relationships between them. Second, it should be noted that while this study investigated the link between basic psychological needs and physical activity, self-efficacy, and resilience, it did not address the issue of whether autonomy, competence, and relatedness needs are independently important in the present setting. As suggested by *Van der Kaap-Deeder et al. (2017)*, each psychological need may contribute uniquely to adolescents' well-being and may play a more distinct role in more specific domains. Therefore, future work should examine the association between each psychological need and the links between physical activity and resilience or self-efficacy. Third, the present study also did not comprehensively estimate the association between the type, duration, intensity, and frequency of physical activity, so further empirical and qualitative work is necessary to determine the effects of these factors.

## CONCLUSIONS

It seems that this is the first study to investigate physical activity, basic psychological needs, self-efficacy, and resilience as an interactive system. Our analysis has revealed a direct and significant correlation between physical activity and resilience among adolescents. We also found that the relationship between physical activity and resilience is influenced by the sequential associations between basic psychological needs and self-efficacy. Regular physical activity can induce positive psychology changes, ultimately contributing to the enhancement of resilience levels. Although further research is needed, our findings suggest that the fulfillment of basic psychological needs and higher levels of self-efficacy may play pivotal roles in improving resilience among adolescents within the context of a physical activity environment.

## ACKNOWLEDGEMENTS

The authors wish to thank the volunteer participants for their valuable time and contribution.

### Funding

The authors received no funding for this work.

### Competing Interests

The authors declare there are no competing interests.

### Author Contributions

- Xuening Li conceived and designed the experiments, performed the experiments, analyzed the data, prepared figures and/or tables, authored or reviewed drafts of the article, and approved the final draft.
- Jing Wang conceived and designed the experiments, analyzed the data, prepared figures and/or tables, authored or reviewed drafts of the article, and approved the final draft.
- Huasen Yu conceived and designed the experiments, performed the experiments, analyzed the data, prepared figures and/or tables, and approved the final draft.
- Yang Liu analyzed the data, authored or reviewed drafts of the article, and approved the final draft.
- Xiaoling Xu performed the experiments, authored or reviewed drafts of the article, and approved the final draft.
- Jiabin Lin analyzed the data, authored or reviewed drafts of the article, and approved the final draft.
- Ning Yang conceived and designed the experiments, authored or reviewed drafts of the article, and approved the final draft.

### Human Ethics

The following information was supplied relating to ethical approvals (*i.e.*, approving body and any reference numbers):

Academic Committee of School of Shandong Youth University of Political Science Special Committee on Scientific Ethics.

### Data Availability

All data are available in the Supplemental File.

### Supplemental Information

Supplemental information for this article can be found online at http://dx.doi.org/10.7717/peerj.17059#supplemental-information.

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
