# Peer review of "How does physical activity improve adolescent resilience? Serial indirect effects via self-efficacy and basic psychological needs"

_PeerJ, doi:10.7717/peerj.17059_

## Round 0.1 · original submission · Major Revisions

Please, attend the reviewers´ comments. Manuscript must be improved.

Best regards

Dr. Manuel Jiménez

**Language Note:** The review process has identified that the English language must be improved. PeerJ can provide language editing services - please contact us at copyediting@peerj.com for pricing (be sure to provide your manuscript number and title). Alternatively, you should make your own arrangements to improve the language quality and provide details in your response letter. – PeerJ Staff

Reviewer 1 ·

Basic reporting

Thanks for providing me the opportunity to review such an interesting manuscript. Some major concerns should be addressed to enhance the manuscript quality:

1. Please, set the goal in the abstract, introduction and discussion. Besides, in the abstract's conclusions it is missed the practical applications of the study.
2. Writing may be improved by erasing a few words in many sentences. Please, revise the manuscript.
3. several statements do not have quotations to support them. For instance, lines 64-65; 296-297.
4. Please, check text citations and references and abide by the journal's rules.
5. In methods, please, separate the "participants and procedure"
6. In the measures, a Cronbach alpha of .65 is too low, according to several authors who claims that alphas should be higher than .70
7. Why you did not separate the factors of the measured variables in the model? I think this could provide more information for the reader.
8. You stated in the introduction and manuscript that physical activity enhances resilience, but I hesitate (e.g., may coach's punishments be positive for resilience?......)

Some more concerns were added to the PDF archive, so you may see them in the attachment.

Experimental design

No comment.

Validity of the findings

No comment.

Additional comments

No comment.

Annotated reviews are not available for download in order to protect the identity of reviewers who chose to remain anonymous.

Reviewer 2 ·

Basic reporting

Paper is written very well. This paper will be very good in the self-efficacy and psychological needs.

Experimental design

Study design is carried out in a systematic way.

Validity of the findings

No comment

Reviewer 3 ·

Basic reporting

The language should be revised by a fluent English speaker. The reviewer suggests that the manuscript be rewritten. For example, the abstract section is not attractive. The ideas are not organized and there are repeated phrases: “This study uses a serial indirect mediator model to explore how physical activity improves adolescent resilience. Methods: This study uses a serial indirect mediator model to explore how physical activity improves adolescent resilience.”

Experimental design

Resilience is a general term that can be used in different context. For example, resilience to climate change, resilience to stress, resilience to environmental adversities. The reviewer suggests highlighting throughout the text “psychological resilience” or another more specific term.

Validity of the findings

The novelty of the study should be better expressed. In discussion section, the authors mention several times “Many studies have demonstrated that adolescent who regularly participate in physical activity exhibit a higher level of self-efficacy” “Taken together, these studies strongly support that self-efficacy plays a role in the buffering effect of physical activity and in how physical activity contributes to resilience”. The difference between previous studies and those in the manuscript needs to be clarify.

---

## Round 0.2 · Minor Revisions

Please, some changes must be made.

Thank you for your patience.

Dr. Manuel Jiménez

Reviewer 1 ·

Basic reporting

Thanks for providing me with the opportunity to review this manuscript again. It is nice to see that the article is improving during the process. Nevertheless, I have a few concerns that are detailed below:
- In the abstract introduction, you should mention something about physical activity, which is vital to understanding the manuscript.
In the abstract, please state the number of boys and girls, mean age, and age range.
- In the abstract. Why the findings of this study are helpful? Please be more specific.
- Line 60-61. Is this a hypothesis? Can you support this with a quotation?
- Line 100. Please divide it into two sentences.
- In the last paragraph of the introduction, you must support the information with quotations.
- In the study design section, it is interesting to set the independent variables and dependent variables

Experimental design

Everything is fine.

Validity of the findings

Everything is fine.

Reviewer 3 ·

Basic reporting

ok

Experimental design

ok

Validity of the findings

ok

---

## Round 0.3 · accepted · Accept

Dear Authors:

Thank you for resubmitting your manuscript. Your paper has just been accepted for publication in the PeerJ Journal.

Congratulations.

Dr. Manuel Jiménez